# Mitochondrial Oxidative Phosphorylation Alterations in Placental Tissues from Early- and Late-Onset Preeclampsia

**DOI:** 10.3390/ijms26093951

**Published:** 2025-04-22

**Authors:** Theresa Lehenauer, Heidi Jaksch-Bogensperger, Sara Huber, Daniel Weghuber, Thorsten Fischer, Johannes A. Mayr, Barbara Kofler, Bettina Neumayer, Daniel Gharehbaghi, Michaela Duggan-Peer, Maximilian Brandstetter, Claudius Fazelnia, René G. Feichtinger

**Affiliations:** 1University Children’s Hospital, Salzburger Landesklinken (SALK) and Paracelsus Medical University (PMU), Müllner Hauptstraße 48, 5020 Salzburg, Austria; t.lehenauer@salk.at (T.L.); d.weghuber@salk.at (D.W.); h.mayr@salk.at (J.A.M.); 2Department of Obstetrics and Gynaecology, Salzburger Landesklinken (SALK) and Paracelsus Medical University (PMU), Müllner Hauptstraße 48, 5020 Salzburg, Austria; h.jaksch-bogensperger@salk.at (H.J.-B.); th.fischer@salk.at (T.F.); da.gharehbaghi@salk.at (D.G.); m.duggan-peer@salk.at (M.D.-P.); m.brandstetter@salk.at (M.B.); c.fazelnia@salk.at (C.F.); 3Research Program for Receptor Biochemistry and Tumor Metabolism, University Children’s Hospital, Salzburger Landesklinken (SALK) and Paracelsus Medical University (PMU), Müllner Hauptstraße 48, 5020 Salzburg, Austria; sar.huber@salk.at (S.H.); b.kofler@salk.at (B.K.); 4Department of Pathology, Salzburger Landesklinken (SALK) and Paracelsus Medical University (PMU), Müllner Hauptstraße 48, 5020 Salzburg, Austria; b.neumayer@salk.at

**Keywords:** preeclampsia, mitochondria, oxidative phosphorylation, placenta

## Abstract

Preeclampsia (PE), a pregnancy complication characterized by high blood pressure and organ damage, has been suggested to be associated with mitochondrial dysfunction, although evidence remains limited. This study aimed to investigate the activity of oxidative phosphorylation (OXPHOS) enzymes and the expression of related proteins in placental tissues from women diagnosed with early-onset preeclampsia (eoPE, <34 weeks of gestation), late-onset preeclampsia (loPE, ≥34 weeks of gestation), and normotensive controls. Placental samples were analyzed using immunohistochemistry, western blotting, and enzymatic activity assays to assess the activity and expression of OXPHOS complexes. Complex I activity was increased by 80% in eoPE and 56% in loPE, with positive correlations between normalized complex I expression, gestational age at delivery (r = 0.85, *p* = 0.01), and birth weight (r = 0.88, *p* = 0.004) in loPE. Relative complex II expression in loPE showed positive correlations with pregnancy duration (r = 0.76, *p* = 0.03) and birth weight (r = 0.77, *p* = 0.03), while in controls, complex II expression correlated with pregnancy duration (r = 0.64, *p* = 0.03). Additionally, complex IV enzyme activity in eoPE was negatively correlated with maternal age at birth (r = −0.69, *p* = 0.03). The observed correlations highlight mitochondrial metabolism as a promising biomarker for predicting disease progression and guiding therapeutic interventions in preeclampsia. Unraveling its precise role in PE pathogenesis is critical to advancing diagnostic precision and improving maternal-fetal outcomes.

## 1. Introduction

The placenta is a temporary organ, which develops during pregnancy and plays a crucial role in ensuring the health and growth of the fetus. The placenta transports oxygen (O_2_), nutrients, ions, and important micronutrients from the mother to the fetus and waste products such as carbon dioxide (CO_2_) in the opposite direction [1]. Proper placental development, which is critical for a healthy pregnancy, depends on trophoblastic invasion and differentiation into cytotrophoblasts and syncytiotrophoblasts, which form villous trees that infiltrate the maternal decidua [2]. Impaired placental development increases oxidative stress, delays fetal growth, and can lead to complications like miscarriage, stillbirth, fetal growth restriction (FGR), and preeclampsia (PE) [3]. PE affects up to 5% of pregnancies [4] and is characterized by new-onset hypertension (>140 mmHg systolic or 90 mmHg diastolic blood pressure) at >20 weeks of gestation, accompanied by proteinuria and other maternal organ dysfunction like liver, brain, or kidneys, and/or FGR [5,6]. For both the mother and the fetus, PE can be lethal, and women who have been affected by PE in the past are predisposed to developing cardiovascular diseases in later life [5]. The only definitive treatment for PE is delivery, often premature, as symptoms typically resolve quickly after birth [7]. Although the exact cause of PE remains unclear, it is widely acknowledged that the placenta, rather than the fetus, is thought to be the primary driver of this disease [5,8,9]. The placenta requires a range of substrates to meet its energy needs and has a higher weight-specific O_2_ consumption than the adult or fetus [10]. This high rate of O_2_ consumption also indicates a marked dependence on oxidative phosphorylation (OXPHOS) for proper placental function. Further evidence of the promising importance of mitochondrial energy metabolism in PE pathogenesis comes from studies of patients with mitochondriopathies, such as mitochondrial encephalomyopathy with lactat acidosis and stroke-like episodes (MELAS) syndrome, which is associated with increased risk of PE [11]. OXPHOS is responsible for supplying cells with energy via the production of adenosine triphosphate (ATP). This process takes place in the inner mitochondrial membrane and is carried out by the electron transport chain (complex I–IV) and complex V, also known as ATP synthase [12].

PE is commonly classified into two categories: early-onset PE (eoPE) and late-onset PE (loPE) [2,5,11]. If symptoms and birth occur before the 34^+0^ week of pregnancy, the condition is classified as eoPE, which is also known as ‘placental PE’ because of its association with abnormal placentation and FGR [6]. Although this subtype occurs in only 5–20% of all PE cases, it is clinically important because it causes the majority of fetal and maternal deaths. Early-onset PE is associated with inadequate development of the cytotrophoblasts, resulting in abnormal growth of the villi. This can also be seen in loPE, but it is often a consequence rather than the primary cause [2,3]. Histomorphologically, no differences between eoPE and loPE are reported. Consequently, the reduced invasion of the trophoblast cells potentially causes a diminished spiral arteria transformation, further lowering the exchange of O_2_ and nutrients [2,12]. The hypoxic environment characterizing PE pregnancies is also the cause of the increased oxidative stress found in PE [13]. The resulting hypoxia stimulates the secretion of anti-angiogenic factors into the maternal blood. This release of vasoactive factors leads to the secretion of inflammatory cytokines and chemokines, which are responsible for the characteristic vascular inflammatory signs of PE, including endothelial dysfunction and hypertension [5]. Late-onset PE is defined as the onset of symptoms and delivery after the 34th week of pregnancy. This subtype accounts for over 80% of all cases of PE and is often termed ‘maternal PE’ because placental dysfunction affects the maternal endothelium but does not appear to affect fetal development [2]. It is the hypothesis that loPE results from chronic systemic inflammation, leading to an imbalance between normal maternal blood flow and placental and fetal metabolic demands [5]. Maternal risk factors include a genetic predisposition, metabolic disorder, and a high body mass index (BMI).

This study aimed to investigate the activity of oxidative phosphorylation (OXPHOS) enzymes and the expression of related proteins in placental tissues from women diagnosed with early-onset preeclampsia (eoPE, <34 weeks of gestation), late-onset preeclampsia (loPE, ≥34 weeks of gestation), and normotensive controls. To our knowledge, this is the first study that uses three complementary approaches to elucidate changes in OXPHOS in PE. In detail, we immunohistochemically (IHC) stained subunits of each of the five OXPHOS complexes to evaluate total differences in protein expression. In addition, we performed western blot (WB) analyses with voltage-dependent anion channel 1 (VDAC1), also known as porin, as loading control to normalize OXPHOS subunit expression to mitochondrial amount. Furthermore, we carried out a spectrophotometric determination of enzymatic activity of the OXPHOS complexes, because in principle it is possible that just the enzymatic activity is affected and not the amount of protein.

## 2. Results

The aerobic mitochondrial energy metabolism was analyzed in placentas from full-term pregnancies (n = 9–13) and PE (n = 16–18). The PE tissues were divided into two groups for further analysis according to the most common subclassification of PE, eoPE (<34 weeks; n = 8–10) and loPE (≥34 weeks; n = 8) and analyzed for mitochondrial content and OXPHOS complexes. Furthermore, the parameters measured were also correlated to clinical parameters. A detailed description of the study cohort is provided in Table 1. No statistical significance was present for age at birth and BMI. CTRL vs. eoPE showed a significant difference in pregnancy duration (*p* < 0.0001) and birth weight (*p* < 0.0001), and eoPE vs. loPE in pregnancy duration (*p* = 0.0086).

The age of the mother at birth is given in years; pregnancy duration is given in weeks + days; BMI: body mass index. CTRL (n = 13), eoPE (n = 10), loPE (n = 8); a non-parametric Kruskal–Wallis test with Dunn’s multiple comparisons test was used to compare the clinical parameters (age at birth, BMI, pregnancy duration, birth weight) of the CTRL, eoPE, and loPE group. Mean ± standard deviation (SD) for every group and parameter is given in the table. No statistical significance was present for age at birth and BMI. CTRL vs. eoPE showed a significant difference in pregnancy duration (*p* < 0.0001) and birth weight (*p* < 0.0001), and eoPE vs. loPE in pregnancy duration (*p* = 0.0086).

### 2.1. Mitochondrial Mass

To elucidate whether mitochondrial mass is changed in PE compared to controls, we used different markers for mitochondrial mass/biogenesis like VDAC1, a mitochondrial outer membrane protein, the citrate synthase in the mitochondrial matrix, and mitochondrial DNA (mtDNA) copy number (Figure 1 and Figure 2). VDAC1 levels in WB (normalized to glyceraldehyde-3-phosphate dehydrogenase (GAPDH)), IHC, as well as the enzymatic activity of the citrate synthase (CS) and mtDNA copy numbers were equal in control tissues compared to eoPE and loPE (Figure 3a–c). The mtDNA copy number was 408 ± 198 copies/cell for control tissues, 497 ± 174 copies/cell for eoPE and 480 ± 343 copies/cell for loPE.

### 2.2. Mitochondrial Complex I (NADH:ubiquinone oxidoreductase)

Although no significant differences in total or relative complex I subunit (NDUFS4) expression were observed (Figure 3d,e), a trend towards higher complex I enzyme activity was observed in both eoPE (+80%) and loPE (+56%) (Figure 3f). In the loPE group, there was also a significant positive correlation between relative NDUFS4/VDAC1 levels and pregnancy days (r = 0.85; *p* = 0.007 Table 2) as well as birth weight (r = 0.88; *p* = 0.004; Table 2).

### 2.3. Mitochondrial Complex II (Succinate dehydrogenase, SDH)

Analysis of total/relative SDHA expression and complex II enzyme activity revealed no difference between eoPE, lope, and control groups (Figure 3g–i). However, there was a significant positive correlation between total SDHA levels on IHC staining and pregnancy duration in the control group (r = 0.683; 0.035; Table 2).

### 2.4. Mitochondrial Complex III (Ubiquinol-cytochrome c reductase)

No significant differences in total/relative UQCRC2 levels or complex III activity were observed between eoPE, lope, and controls (Figure 3j–l). However, there was a significant inverse correlation between total UQCRC2 expression and birth weight (r = −0.776; *p* = 0.024; Table 2).

### 2.5. Mitochondrial Complex IV (Cytochrome c oxidase, COX)

Neither changes in total/relative MT-CO2 levels nor complex IV activity were present in eoPE and loPE (Figure 3m–o). A significant inverse correlation between complex IV/CS activity and age at birth was observed only in eoPE (r = −0.0695; *p* = 0.026; Table 2).

### 2.6. Mitochondrial Complex V (ATP synthase)

There were no differences between eoPE, lope, and controls regarding complex V protein levels or activity (Figure 3p–r). Furthermore, there were no correlations involving this complex identified.

## 3. Discussion

This study did not identify any significant differences between controls, eoPE, and loPE in relation to the mitochondrial amount as ascertained by three different methodologies: IHC, WB, and spectrophotometry (Figure 3). Our findings are in agreement with a previous study that shows equivalent VDAC1 expression in controls, eoPE, and loPE [14]. In contrast, in the same study, a significant higher in CS activity in loPE and eoPE and mtDNA copy number in eoPE was reported compared to controls [14]. The exact values for the median mtDNA and CS were not specified, but the differences between the groups were minor. In contrast, two other PE studies reported a significantly lower mtDNA copy number and CS activity [7,15], arguing for a diminished mitochondrial mass. Surprisingly, the O_2_ flux via complex I and complex I + II was higher in those studies [7]. Consistent with the diminished CS activity, the authors found a downregulation of all complexes in WB in the pre-term groups. Due to analyzing the total and relative mitochondrial amount, using three different methods, our study could rule out any significant or pathologically relevant changes in mitochondrial biogenesis during pregnancy.

One relatively consistent finding is an increase in complex I activity and O_2_ flux via complex I in PE. In keeping with previous results, a notable trend towards higher complex I activity with approximate increases of 80% and 56% in eoPE and loPE compared to the control group was observed in our study (Figure 3). The increase in O_2_ flux via complex I, as previously described by Holland et al., is also in agreement with our results [15], as is a study from 2016, which reported the phenomenon for eoPE [2,16]. On the contrary, a recent study from Vaka et al. reported a downregulation of O_2_ flux [17]. Complex I is a giant multisubunit protein complex encoded by both the nuclear and mtDNA further organized in supercomplexes with complex III and IV as well as accessory subunits whose assembly is guided by a complex assembly machinery. Due to its complexity, it is highly vulnerable to damage, which explains why it is often affected by disorders of energy metabolism [18,19]. Importantly, complex I is also involved in several key aspects of cell biology, such as fatty acid oxidation and apoptosis [20,21]. Increased complex I activity may potentially affect the NAD^+^/NADH ratio. Jahan et al. [22] reported an increased activity of NAD^+^-consuming enzymes and decreased NAD^+^ content in inflammation-driven PE in a rat model. Nicotinamide riboside administration prevented maternal hypertension and FGR/placental growth restriction [22]. Another study, using mouse PE models, also showed that mothers and pups benefited from nicotinamide [23]. Since complex I should principally elevate the NAD^+^/NADH ratio, a high complex I activity, as found in our study, might be beneficial. Therefore, the increased complex I activity might not be a sign of a complex I defect but rather represents a compensatory mechanism to fulfill the increased NAD^+^ demands present in PE. NAD^+^, NAD precursors, and NAD analogues might therefore represent promising compounds to treat PE.

Antiphospholipid syndrome (APS) is a systemic autoimmune disease characterized by the presence of antiphospholipid antibodies. PE occurs in approximately 10–17% of pregnancies with APS. Only eoPE belongs to the clinical criteria of APS. Similarities in the pathophysiology of eoPE and APS emphasize an association of these two syndromes. A diminished trophoblast invasion and eoPE are frequently observed in APS [24,25]. The high number of anticardiolipin-IgG positive centenarians of 21% without known autoimmune disease [26] suggests that cardiolipin metabolism might play a bigger role in PE than previously thought. Cardiolipin is an important component of the inner mitochondrial membrane. Barth syndrome is caused by variants in tafazzin, an enzyme involved in cardiolipin synthesis exhibiting complex I deficiency due to assembly defects [27]. The activity of respiratory complex I is greatly increased by cardiolipin [28].

As complex I oxidizes NADH generated during β-oxidation, complex I contributes to the efficient generation of ATP from fatty acids [12]. Yu et al. [29] observed impaired long-chain fatty acid oxidation in trophoblasts treated with serum from patients with severe eoPE, severe eoPE with hemolysis, elevated liver enzymes, and low platelet count (HELLP) syndrome and APS. This treatment resulted in altered mitochondrial morphology and increased lipid droplet deposition, similar to the changes observed in trophoblasts exposed directly to long-chain fatty acids [2,29], suggesting a potential dysfunction in long-chain fatty acid oxidation in eoPE. This impairment may be due to the observed reduced fatty acid oxidation in placentas affected by PE [30,31]. This aligns with the findings of Yu et al. [29] and suggests that circulating factors in eoPE may disrupt placental fatty acid metabolism, leading to compensatory changes in mitochondrial function via increase of complex I activity.

Apart from energy production, another major function of complex I is apoptosis. The p75 (NDUFS1) subunit can be cleaved by caspase 3 [32] and granzyme A cleaves NDUFS3, subunit to initiate caspase-independent cell death [33,34]. Granzyme A is present on cytotoxic T lymphocytes [35]. PE is associated with increased cytotoxic T-cell capacity to paternal antigens [36]. Thus, the pathogenic role of increased complex I activity might be independent of its function in respiration because of its pleiotropic nature.

A correlation between the complex I/VDAC1 ratio and both gestational age as well as birth weight was observed (Table 2). Increased complex I expression, relative to overall mitochondrial content, was associated with a better outcome for the fetus due to longer gestation and higher birth weight in loPE. Since the enzyme activity was also altered between loPE, eoPE, and controls, complex I might indeed play a role in PE pathogenesis.

An inverse correlation was present between total complex III levels and birth weight in loPE. Since the correlation with complex I is direct and the correlation with complex III is indirect, complex I and complex III might be regulated independently and not on the supercomplex level. Complex III catalyzes a reaction with coenzyme Q and cytochrome c. Coenzyme Q was previously linked to PE in numerous studies and coenzyme Q supplementation helped prevent the development of PE [37,38]. Complex III is important for immune regulation and T-cell function as well as hypoxia [39,40].

We observed, for instance, a positive correlation between complex II levels and gestational age in both the control group (total complex II levels with IHC) and loPE (relative complex II level with WB) (Table 2). Additionally, there was a significant positive correlation between complex II levels and birth weight in the loPE group (relative complex II level with WB). In principle, a higher complex II amount is associated with a positive outcome in controls and loPE. This is consistent with the findings from Holland et al. [15], who reported that the complex II protein levels are influenced by gestation and who observed increased protein levels in term PE placentas. Complex II (succinate dehydrogenase) is the only complex that is exclusively encoded by the nuclear genome; it is the smallest OXPHOS complex comprising only four subunits and also has a dual role in the citric acid cycle. So far, there is no evidence that succinate or fumarate play a role in PE pathogenesis. However, both fumarate and succinate have an impact on the inflammation process and hypoxia, two factors involved in PE development [41,42,43,44,45]. Overall, these findings suggest that complex II may be upregulated throughout pregnancy in normal and loPE placentas and this upregulation may be associated with improved pregnancy outcomes, as indicated by longer gestation and higher birth weights in the loPE group. However, further research is needed to explain the underlying mechanisms connecting the complex II expression to disease outcomes. One pitfall of the study is the lack of a control group with the same gestational age as the eoPE. Since there is no indication to deliver a healthy child prematurely and there is always a pathology behind it, such samples cannot be obtained.

The inverse correlation between complex IV activity and age at birth in eoPE aligns with the existing literature examining mitochondrial function in aging cells [46,47]. Hypoxia-inducible factor 1-alpha (HIF1A)-mediated silencing of COX5B, a subunit of complex IV, impairs complex IV activity and leads to decreased fatty acid oxidation [47]. The reduced complex IV activity in eoPE placentas, as observed in our study, may further increase the existing impairment of fatty acid metabolism in these pregnancies. This potential interplay between complex IV activity and fatty acid metabolism warrants further investigation as it may contribute to the overall pathophysiology of eoPE or aging in general, as the egg cells, from which the placenta develops, age with the woman and are therefore potentially affected by mitochondrial changes during aging [48,49,50,51].

## 4. Materials and Methods

### 4.1. Sample Collection

Placental tissue samples from female patients with and without PE were obtained from the University Clinics of Gynaecology and Obstetrics, Salzburger Landesklinken (SALK) and Paracelsus Medical University (PMU), Austria. The collected clinical parameters for all individuals in the study cohort are provided in Table 1. Beforehand, the project was approved by the Salzburg Ethics Committee (415-E/2333/9-2018) and every participant signed a written informed consent outlining the purpose of the study and the planned intervention. PE was diagnosed according to international criteria of systolic blood pressure ≥ 140 mmHg and/or diastolic blood pressure ≥ 90 mmHg and the presence of proteinuria (≥0.3 g/24 h or ≥2+ on dipstick analysis) occurring > 20 weeks of gestation in previously normotensive women or the development of proteinuria > 20 weeks of gestation in women with pre-existing hypertension. To ensure that vaginal birth did not affect the results, fresh human placental tissue samples from the maternal side were collected after cesarean section and frozen within 15 min at −80 °C. For IHC analysis, tissues were formalin-fixed and paraffin-embedded (FFPE). For WB analysis and determination of OXPHOS enzyme activity, placental tissue samples were homogenized and centrifuged at 600× *g* for 10 min at 4 °C, as described previously [52]. The supernatant was aliquoted, instantly deep frozen in liquid nitrogen, and stored at −80 °C. For the determination of the mitochondrial copy number, the remaining cell pellet of the lysate preparation was lysed with proteinase K buffer using the following program on the T Personal Thermocycler (Biometra GmbH; Göttingen, Germany): 60 °C for 60 min, 95 °C for 10 min, and 25 °C for 10 s.

The OXPHOS system was examined via three methods: WB [53,54], IHC [54,55], and spectrophotometric determination of OXPHOS enzyme activity [53,54]. The antibodies used for IHC and WB are all suitable to detect OXPHOS assembly defects, since they are incorporated during a late stage of OXPHOS complex assembly. All antibodies have been previously referenced in numerous publications relating to their use in the identification of patients with OXPHOS enzyme defects [53,55,56,57,58].

### 4.2. Immunohistochemistry

FFPE slides (4 µm thick) of placenta tissues were IHC-stained with antibodies against subunits of each of the five OXPHOS complexes and VDAC1. IHC was performed as previously described [59,60] using the Dako Envision Detection System with Peroxidase-Blocking Solution (Agilent Technologies; Inc., Santa Clara, CA, USA; S2023), DAB+Substrate Chromogen (Agilent Technologies; Inc., Santa Clara, CA, USA; K3468), EnVision+ System− HRP Labelled Polymer Anti-mouse (Agilent Technologies; Inc., Santa Clara, CA, USA; K4001). All the antibodies were diluted in Dako Antibody Diluent with Background-Reducing Components (Agilent Technologies; Inc., Santa Clara, CA, USA; S3022). The following antibodies were used: mouse monoclonal anti-VDAC1 (1:1000; Abcam; Cambrige, UK; Ab14734); mouse monoclonal anti-nicotinamide adenine dinucleotide hydrogen (NADH): oxidoreductase subunit S4 (NDUFS4) (1:500; Sigma-Aldrich/Merck KGaA; Darmstadt, Germany; WH0004724M1); mouse monoclonal anti-succinate dehydrogenase complex flavoprotein subunit A (SDHA) (1:1000; Abcam; Cambrige, UK; Ab14715); mouse monoclonal anti-ubiquinol-cytochrome c reductase core protein 2 (UQCRC2) (1:1000; Abcam; Cambrige, UK; Ab14745); mouse monoclonal anti-mitochondrially encoded cytochrome c oxidase I subunit (MT-CO1) (1:1000; Abcam; Cambrige, UK; Ab14705); and mouse monoclonal anti-ATP synthase F1 subunit A (ATP5F1A) (1:1000; Abcam; Cambrige, UK; Ab14748). The slides were incubated with the respective antibody solutions for 1 h at room temperature. The staining intensity was analyzed using a score system from 0 to 3: no staining = 0; weak staining = 1; moderate staining = 2; and strong staining = 3 (Figure 1).

### 4.3. Western Blot

Protein levels of subunits of each of the five OXPHOS complexes were determined by WB. Voltage-dependent anion channel 1 (VDAC1), a marker of mitochondrial mass, and GAPDH were used as loading controls. Thirty (30) μg protein of each sample was loaded on a 10% polyacrylamide gel. Blotting was carried out with the Trans-Blot Turbo device (Bio-Rad Laboratories; Hercules, CA, USA) using the Trans-Blot Turbo Transfer Pack (Bio-Rad Laboratories; Hercules, CA, USA; #1704158). Blots were incubated first with with 10% Western Blocking Reagent (Roche Diagnostics; Mannheim, Germany; 11921673001) an then with the following primary antibodies and incubation times: mouse monoclonal anti-VDAC1 (1:1000; 10 h; 4 °C; Abcam; Cambridge, UK; Ab14734), rabbit polyclonal GAPDH (1:1000; 11 h; 4 °C; Novus Biologicals; Centennial, CO, USA; 2275/PC/100), mouse monoclonal anti-NDUFS4 (1:1000; 2 h; room temperature; Abcam; Cambridge, UK; WH0004724M1), mouse monoclonal anti-SDHA (1:3000; 1 h; room temperature; Abcam; Cambridge, UK; Ab14715), mouse monoclonal anti-UQCRC2 (1:1000; 1 h; room temperature; Abcam; Cambridge, UK; Ab14745), rabbit monoclonal anti-MT-CO2 (1:5000; 15 h; 4 °C; Abcam; Cambridge, UK; Ab79393), and mouse monoclonal anti-ATP5F1A (1: 1000; 1 h; room temperature; Abcam; Cambridge, UK; Ab14748). After washing, the blots were incubated for 2 h with the EnVision+ System− HRP Labelled Polymer Anti-mouse (1:100; Agilent Technologies; Inc., Santa Clara, CA, USA; K4001) for monoclonal mouse antibodies and the EnVision+ System− HRP Labelled Polymer Anti-rabbit (1:100; Agilent Technologies; Inc., Santa Clara, CA, USA; K4003) for the monoclonal rabbit antibodies, at room temperature. Bands were visualized by chemiluminescence using the Lumi-Light Plus Western Blotting Substrate (Roche Diagnostics; Mannheim, Germany; 12015196001) and the Molecular Imager Chemi Doc XRS (Bio-Rad Laboratories; Hercules, CA, USA). Images were analyzed using the Image Lab software version 6.0.1 (Bio-Rad Laboratories; Hercules, CA. USA).

### 4.4. OXPHOS Enzyme Activity

The OXPHOS enzyme activities were determined in post-nuclear placental lysates from approximately 60 mg placenta tissue. Spectrophotometric measurement of the OXPHOS enzyme and CS activity was performed as previously described [52,61,62,63]. For measurement, the UV-Vis-Spectrophotometer Specord 200 plus (Analytik Jena GmbH; Jena, Germany) was used with a continuous water pump and a water bath to maintain a temperature of 37 °C during the whole analysis.

### 4.5. mtDNA Copy Number

For the evaluation of the mtDNA copy number, proteinase K-digested cell pellets from lysate preparations were used. The analysis was performed as previously described [64]. Briefly, mtDNA was amplified by using two target sequences on the mtDNA and four target sequences in nuclear genes. To calculate the mtDNA content in the placenta samples, the ΔCt between the nuclear fragment and the mitochondrial fragment was determined. 2^ΔCt^ gives the mtDNA copy number.

### 4.6. Statistical Analysis

Data were analyzed using Prism6 software version 10.1.2 (GraphPad Software, Boston, MA, USA). The Student’s *t*-test and non-parametric Mann–Whitney U-test were used to compare eoPE and loPE with the control group. To determine correlations between OXPHOS parameters and clinical parameters, the Pearson test was used. Significance was set at *p* < 0.05.

## 5. Conclusions

In summary, the data regarding the OXPHOS system are sparse and often contradictory. In this study, we were able to contribute relevant data about the role of the mitochondrial respiratory system to the pathogenesis of PE. Although the sample size was rather small and this represents a pilot study, we can clarify open questions concerning energy metabolism in PE because of a multi-method approach, determining the total and relative OXPHOS subunit expression, as well as enzyme activity. We hypothesize that complex I might play a role in PE pathogenesis, although potentially it is not defective but dysregulated and contributes to disease severity. Importantly, we highlight interesting correlations between OXPHOS components and clinical parameters, which merit further exploration for their potential use in disease monitoring and predicting PE outcomes. Furthermore, a focus on specific complexes and their role in fatty acid metabolism, NAD^+^/NADH ratios, and apoptosis could offer novel therapeutic strategies, such as NAD^+^ precursors or compounds aimed at enhancing mitochondrial function.

## Figures and Tables

**Figure 1 ijms-26-03951-f001:**
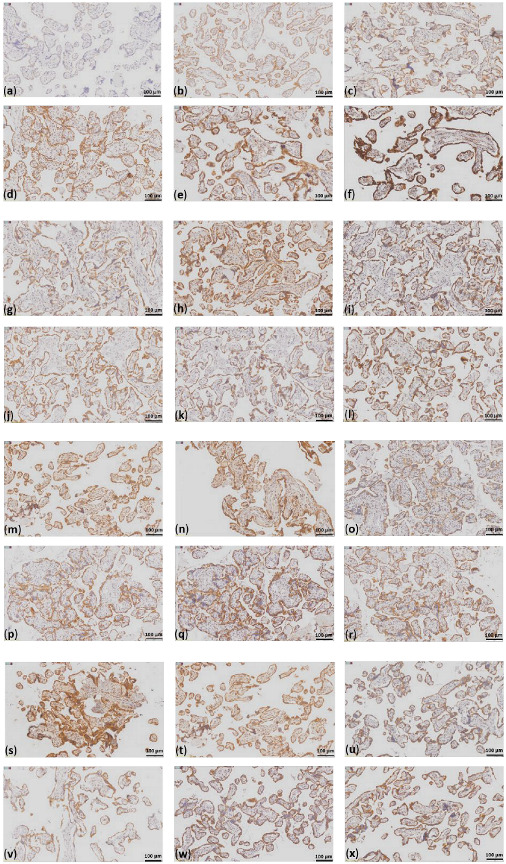
Staining intensities for the OXPHOS subunits and staining of VDAC1 and the OXPHOS subunits in a representative case of each group. (**a**) no to weak staining, 0.5; (**b**) weak staining, 1; (**c**) weak to moderate staining. 1.5; (**d**) moderate staining, 2; (**e**) moderate to strong staining, 2.5; (**f**) strong staining, 3. In the image, different antibody stainings are shown, which most accurately reflect the varying staining intensities; case CTRL 3 staining of (**g**) VDAC1; (**h**) NDUFS4; (**i**) SDHA; (**j**) UQCRC2; (**k**) MT-CO1; (**l**) ATP5F1A; case eoPE 2 staining of (**m**) VDAC1; (**n**) NDUFS4; (**o**) SDHA; (**p**) UQCRC2; (**q**) MT-CO1; (**r**) ATP5F1A; case lope 1 staining of (**s**) VDAC1; (**t**) NDUFS4; (**u**) SDHA; (**v**) UQCRC2; (**w**) MT-CO1; (**x**) ATP5F1A; scale bar = 100 µm.

**Figure 2 ijms-26-03951-f002:**
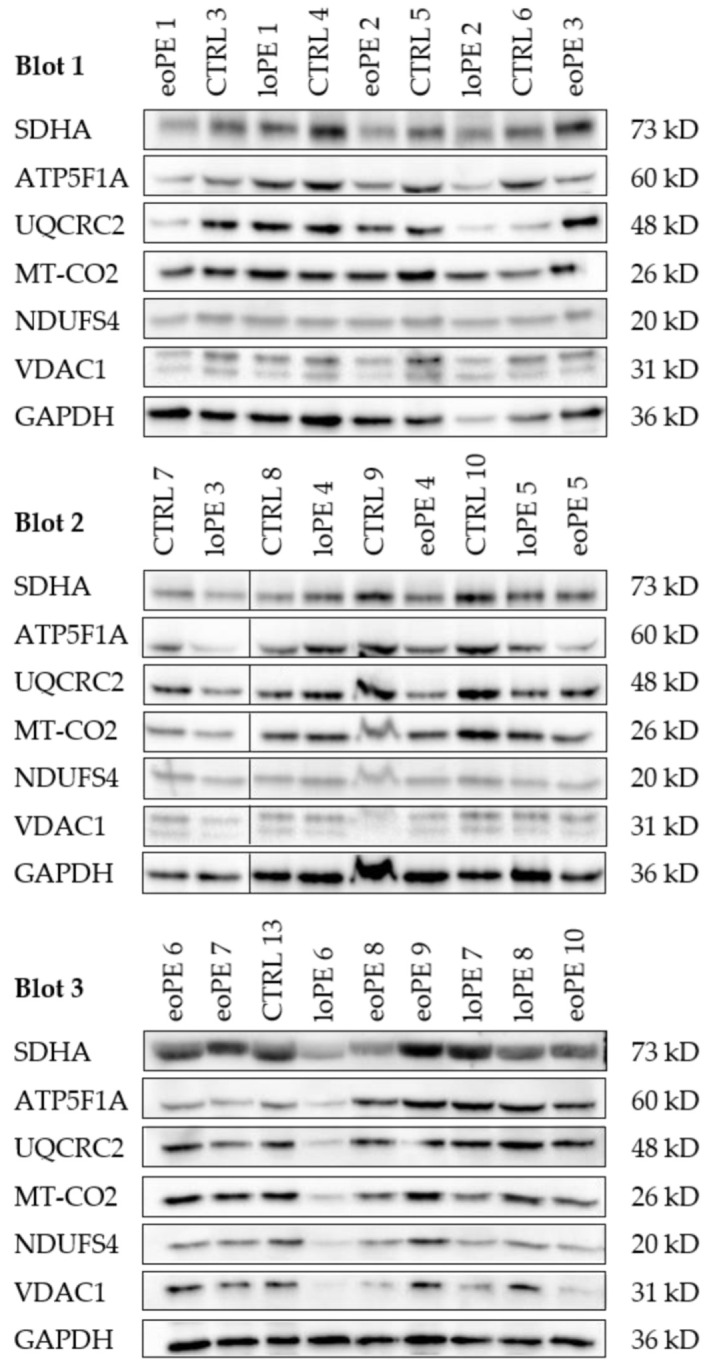
Western blot of subunits of each of the five OXPHOS complexes, VDAC1 and GAPDH. Western blot of each analyzed sample. The samples are described in Table 1. Cytosolic GAPDH and mitochondrial VDAC1 were used as loading controls. The black line after lane two in blot 2 indicates that here the marker was cutted out. In blot 1 and 3 the marker was on left side. Densitometric analysis of the western blots is given in Figure 3.

**Figure 3 ijms-26-03951-f003:**
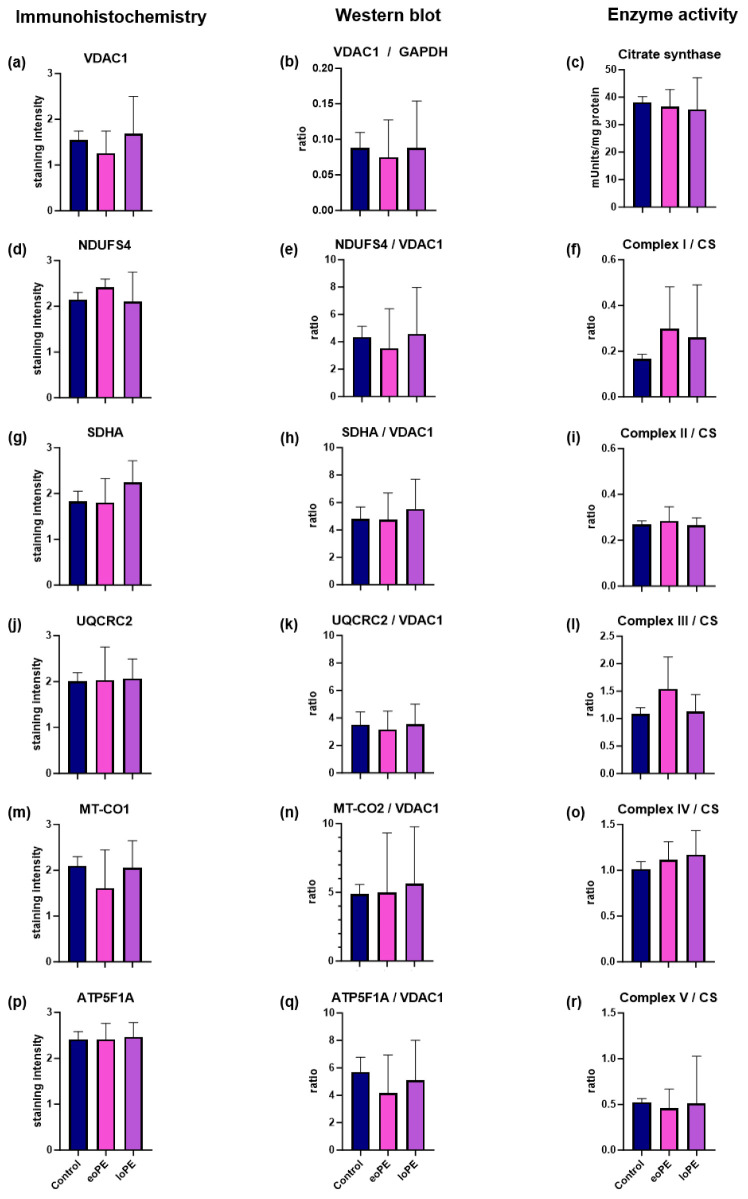
Comparison of total and relative amounts of OXPHOS subunits and OXPHOS enzyme activity in placentas of eoPE, lope, and controls. (**a**,**d**,**g**,**j**,**m**,**p**) IHC staining of VDAC1, NDUFS4 (complex I), SDHA (complex II), UQCRC2 (complex III), MT-CO1 (complex IV), and ATP5F1A (complex V) in placentas from controls, eoPE and loPE pregnancies; (**b**) WB analysis of VDAC1 levels normalized to GAPDH as a loading control; (**e**,**h**,**k**,**n**,**q**) WB analysis of NDUFS4 (complex I), SDHA (complex II), UQCRC2 (complex III), MT-CO2 (complex IV), and AT5F1A (complex V) normalized to VDAC1 as a mitochondrial loading control in controls, eoPE and loPE; (**c**) absolute activity of CS as a marker for the mitochondrial mass; (**f**,**i**,**l**,**o**,**r**) enzyme activities of complex I, II, III, IV, and V normalized to CS in eoPE, lope, and controls. Data are presented as mean ± standard deviation.

**Table 1 ijms-26-03951-t001:** Description of the study cohort.

Sample Number	Group	Age at Birth	BMI	Pregnancy Duration	Number of Pregnancies	Number of Births	Birth Weight
CTRL 1	CTRL	30	31.6	38 + 6	1	1	3510
CTRL 2	CTRL	37	24.9	39 + 2	2	2	3850
CTRL 3	CTRL	26	34	39	2	2	3890
CTRL 4	CTRL	39	25	39	3	2	3320
CTRL 5	CTRL	34	29.4	38	3	3	3210
CTRL 6	CTRL	23	29.7	39 + 5	2	1	3760
CTRL 7	CTRL	32	27	39	2	2	3090
CTRL 8	CTRL	30	27,3	38 + 5	3	3	3460
CTRL 9	CTRL	31	27.9	38 + 4	1	1	3300
CTRL 10	CTRL	35	26.1	39	4	2	3680
CTRL 11	CTRL	39	31.6	39	1	1	3260
CTRL 12	CTRL	30	26.1	39	2	2	3320
CTRL 13	CTRL	33	26,3	38 + 6	3	3	4040
mean ± SD	-	32.23 ± 4.71	28.22 ± 2.83	38 + 6.5 ± 0.39	-	-	3515 ± 300
eoPE 1	eoPE	35	26.1	26	1	1	633
eoPE 2	eoPE	28	35	32 + 6	2	2	2008
eoPE 3	eoPE	30	40.4	30 + 3	1	1	1177
eoPE 4	eoPE	41	NA	32	1	1	1018
eoPE 5	eoPE	37	21.6	30 + 6	6	3	699
eoPE 6	eoPE	24	22.4	32 + 5	1	1	1300
eoPE 7	eoPE	22	27.2	27 + 1	1	1	850
eoPE 8	eoPE	36	52.9	33 + 5	5	3	1970
eoPE 9	eoPE	30	23.9	29 + 1	2	1	950
eoPE 10	eoPE	32	23.6	23 + 4	1	1	355
mean ± SD	-	31.5 ± 5.93	30.34 ± 10.52	29 + 5.9 ± 3.34	-	-	1096 ± 542.9
loPE 1	loPE	29	40.1	39	1	1	2690
loPE 2	loPE	34	NA	38 + 1	1	1	2870
loPE 3	loPE	34	NA	40 + 5	1	1	3540
loPE 4	loPE	28	32.3	39	1	1	3530
loPE 5	loPE	31	45.2	34	1	1	1164
loPE 6	loPE	35	32	34	1	1	1620
loPE 7	loPE	47	32	36 + 5	1	1	1820
loPE 8	loPE	25	30.4	34	2	2	1150
mean ± SD	-	32.88 ± 6.66	35.33 ± 5.93	34 + 6.6 ± 2.68	-	-	2298 ± 988

**Table 2 ijms-26-03951-t002:** Significant correlations between the expression and activity of OXPHOS and clinical parameters.

Correlation	Method	r-Value	*p*-Value	n
**Controls**				
Complex II vs pregnancy duration	**IHC**	**0.638**	**0.035**	**11**
	WB/VDAC1	0.009	0.982	9
	Enzyme activity/CS	0.212	0.509	12
**eoPE**				
Complex IV vs age at birth	IHC	0.542	0.166	8
	WB/VDAC1	0.327	0.356	10
	**Enzyme activity/CS**	**−0.695**	**0.026**	**10**
**loPE**				
Complex I vs pregnancy duration	IHC	−0.326	0.430	8
	**WB/VDAC1**	**0.854**	**0.007**	**8**
	Enzyme activity/CS	−0.556	0.153	8
Complex I vs birth weight	IHC	−0.336	0.416	8
	**WB/VDAC1**	**0.876**	**0.004**	**8**
	Enzyme activity/CS	−0.615	0.104	8
Complex II vs pregnancy duration	IHC	0.023	0.958	8
	**WB/VDAC1**	**0.757**	**0.030**	**8**
	Enzyme activity/CS	0.179	0.671	8
Complex II vs birth weight	IHC	−0.076	0.858	8
	**WB/VDAC1**	**0.766**	**0.027**	**8**
	Enzyme activity/CS	0.022	0.959	8
Complex III vs birth weight	**IHC**	**−0.776**	**0.024**	**8**
	WB/VDAC1	0.384	0.347	8
	Enzyme activity/CS	−0.351	0.394	8

Abbreviations: citrate synthase (CS), immunohistochemistry (IHC), voltage-dependent anion channel 1 (VDAC1), western blot (WB), early-onset preeclampsia (eoPE), late-onset preeclampsia (loPE). Significant correlations are highlighted in bold.

## Data Availability

The original data are deposited at a local server and can be shared upon request.

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
