# Peer review of "Mitochondrial Oxidative Phosphorylation Alterations in Placental Tissues from Early- and Late-Onset Preeclampsia"

_ijms, 2025, doi:10.3390/ijms26093951_

Round 1
Reviewer 1 Report
Comments and Suggestions for Authors
The authors provide interesting data on the role of the mitochondrial respiratory system in the pathogenesis of preeclampsia, a pregnancy-specific syndrome with multisystem involvement leading to fetal, neonatal, and maternal morbidity and mortality, making this a relevant study for the development of targeted therapies. Some suggestions are made below:
The title of the manuscript could be more specific, for example: "Mitochondrial Oxidative Phosphorylation Alterations in Placental Tissues from Early- and Late-Onset Preeclampsia"
In the abstract, avoid the use of unnecessary abbreviations, for example IHC or WB.
Check line 25 "pregnany".
Improve the wording of the objective in the abstract: “This study aimed to investigate the activity of oxidative phosphorylation (OXPHOS) enzymes and the expression of related proteins in placental tissues from women diagnosed with early-onset preeclampsia (eoPE, <34 weeks of gestation), late-onset preeclampsia (loPE, ≥34 weeks of gestation), and normotensive controls.”
Improve the conclusion in the abstract, for example: "The observed correlations highlight mitochondrial activity as a promising biomarker for predicting disease progression and guiding therapeutic interventions in preeclampsia. Unraveling its precise role in PE pathogenesis is critical to advancing diagnostic precision and improving maternal-fetal outcomes."
Include the objective in the Introduction.
Include brands and countries of reagents and antibodies used.
What is the version of GraphPad?
You could use the following manuscript in your
The authors provide interesting data on the role of the mitochondrial respiratory system in the pathogenesis of preeclampsia, a pregnancy-specific syndrome with multisystem involvement leading to fetal, neonatal, and maternal morbidity and mortality, making this a relevant study for the development of targeted therapies. Some suggestions are made below:
The title of the manuscript could be more specific, for example: "Mitochondrial Oxidative Phosphorylation Alterations in Placental Tissues from Early- and Late-Onset Preeclampsia"
In the abstract, avoid the use of unnecessary abbreviations, for example IHC or WB. Check line 25 "pregnany".
Improve the wording of the objective in the abstract: “This study aimed to investigate the activity of oxidative phosphorylation (OXPHOS) enzymes and the expression of related proteins in placental tissues from women diagnosed with early-onset preeclampsia (eoPE, <34 weeks of gestation), late-onset preeclampsia (loPE, ≥34 weeks of gestation), and normotensive controls.”
Improve the conclusion in the abstract, for example: "The observed correlations highlight mitochondrial activity as a promising biomarker for predicting disease progression and guiding therapeutic interventions in preeclampsia. Unraveling its precise role in PE pathogenesis is critical to advancing diagnostic precision and improving maternal-fetal outcomes."
Include the objective in the Introduction.
Include brands and countries of reagents and antibodies used.
What is the version of GraphPad?
You could use the following manuscript in your
The authors provide interesting data on the role of the mitochondrial respiratory system in the pathogenesis of preeclampsia, a pregnancy-specific syndrome with multisystem involvement leading to fetal, neonatal, and maternal morbidity and mortality, making this a relevant study for the development of targeted therapies. Some suggestions are made below:
The title of the manuscript could be more specific, for example: "Mitochondrial Oxidative Phosphorylation Alterations in Placental Tissues from Early- and Late-Onset Preeclampsia"
In the abstract, avoid the use of unnecessary abbreviations, for example IHC or WB. Check line 25 "pregnany".
Improve the wording of the objective in the abstract: “This study aimed to investigate the activity of oxidative phosphorylation (OXPHOS) enzymes and the expression of related proteins in placental tissues from women diagnosed with early-onset preeclampsia (eoPE, <34 weeks of gestation), late-onset preeclampsia (loPE, ≥34 weeks of gestation), and normotensive controls.”
Improve the conclusion in the abstract, for example: "The observed correlations highlight mitochondrial activity as a promising biomarker for predicting disease progression and guiding therapeutic interventions in preeclampsia. Unraveling its precise role in PE pathogenesis is critical to advancing diagnostic precision and improving maternal-fetal outcomes."
Include the objective in the Introduction.
Include brands and countries of reagents and antibodies used.
What is the version of GraphPad?
You could use the following manuscript in your discussion: Marín, R., Chiarello, D. I., Abad, C., Rojas, D., Toledo, F., & Sobrevia, L. (2020). Oxidative stress and mitochondrial dysfunction in early-onset and late-onset preeclampsia. Biochimica et biophysica acta. Molecular basis of disease, 1866(12), 165961. https://doi.org/10.1016/j.bbadis.2020.165961
Author Response
We also added a version with tracked changes to make it easier for the reviewers.
Reviewer 1 - Comment 1:
The title of the manuscript could be more specific, for example: "Mitochondrial Oxidative Phosphorylation Alterations in Placental Tissues from Early- and Late-Onset Preeclampsia."
Reply to Reviewer 1 – Comment 1:
We have changed the manuscript title according to the reviewer’s suggestion.
Reviewer 1 - Comment 2:
In the abstract, avoid the use of unnecessary abbreviations, for example IHC or WB.
Reply to Reviewer 1 – Comment 2:
We have removed the unnecessary abbreviations.
Reviewer 1 - Comment 3:
Check line 25 "pregnany."
Reply to Reviewer 1 – Comment 3:
The typo has been corrected.
Reviewer 1 - Comment 4:
Improve the wording of the objective in the abstract: “This study aimed to investigate the activity of oxidative phosphorylation (OXPHOS) enzymes and the expression of related proteins in placental tissues from women diagnosed with early-onset preeclampsia (eoPE, <34 weeks of gestation), late-onset preeclampsia (loPE, ≥34 weeks of gestation), and normotensive controls.”
Reply to Reviewer 1 – Comment 4:
We appreciate the improvement of the wording and have changed the text in the abstract according to the reviewer’s suggestion.
Reviewer 1 - Comment 5:
Improve the conclusion in the abstract, for example: "The observed correlations highlight mitochondrial activity as a promising biomarker for predicting disease progression and guiding therapeutic interventions in preeclampsia. Unraveling its precise role in PE pathogenesis is critical to advancing diagnostic precision and improving maternal-fetal outcomes."
Reply to Reviewer 1 – Comment 5:
The wording of the conclusion has been changed according to the reviewer’s suggestion.
Reviewer 1 - Comment 6:
Include the objective in the Introduction.
Reply to Reviewer 1 – Comment 6:
In response to the reviewer’s suggestion, we have included the objective as kindly provided by the reviewer in the introduction:
“This study aimed to investigate the activity of oxidative phosphorylation (OXPHOS) enzymes and the expression of related proteins in placental tissues from women diagnosed with early-onset preeclampsia (eoPE, <34 weeks of gestation), late-onset preeclampsia (loPE, ≥34 weeks of gestation), and normotensive controls.”
Reviewer 1 - Comment 7:
Include brands and countries of reagents and antibodies used.
Reply to Reviewer 1 – Comment 7:
The brands/countries of the antibodies have been included.
Reviewer 1 - Comment 8:
What is the version of GraphPad?
Reply to Reviewer 1 – Comment 8:
The information that GraphPad Prism 10.1.2 was used has been included.
Reviewer 1 - Comment 9:
You could use the following manuscript in your discussion: Marín, R., Chiarello, D. I., Abad, C., Rojas, D., Toledo, F., & Sobrevia, L. (2020). Oxidative stress and mitochondrial dysfunction in early-onset and late-onset preeclampsia. Biochimica et biophysica acta. Molecular basis of disease, 1866(12), 165961. https://doi.org/10.1016/j.bbadis.2020.165961
Reply to Reviewer 1 – Comment 9:
According to the reviewer, we have included relevant informations from the mentioned publication also in the discussion.

Reviewer 2 Report
Comments and Suggestions for Authors
In this manuscript authors investigating the oxidative phosphorylation (OXPHOS) enzyme activity and related protein expression in placental tissues from women with early- and late-onset PE, and controls. Authors found that th Complex I activity was increased by both early and late PE. Relative complex II expression in loPE showed positive correlations with pregnancy duration and birth weight, while in controls, complex II expression correlated with pregnany duration. Furthermore, authors found that complex IV enzyme activity in eoPE was negatively correlated with maternal age at birth.
although the manuscript is interesting and generally well written, it presents several issues that must be resolved. In particular:
Lines 69-71: it deserves to be pointed out that the hypoxic environment charactherizing PE pregnancies is also the cause of an increased oxidative stress found in PE (see PMID: 39456486)
2. Materials and Methods: Authors must add the product code of all reagents and kits used in order to allow data reproducibility
Figure 2, Immunohistochemistry: Representative images must be shown
Figure 2, Western blots: Representative images must be shown
Supplementary Table 1: this table should be moved in the manuscript and statistically significant differences of the parameters (Age, BMI, Gestational age at delivery...) among the groups must be reported. In addition, in the eoPE group the gestational age is very lower than the control group, this may significantly alter the results obtained by the authors since the expression of the proteins analysed can be dependent on the physiological gestation age rather than the pathology. Thus, an appropriate control group is necessary.
It is impressive that such a simple study required 13 authors. A detailed author contribution section is necessary.
Author Response
Reviewer 2:
Reviewer 2 - Comment 1:
Lines 69-71: it deserves to be pointed out that the hypoxic environment characterizing PE pregnancies is also the cause of an increased oxidative stress found in PE (see PMID: 39456486).
Reply to Reviewer 2 – Comment 1:
We have included the mentioned statement and reference.
Reviewer 2 - Comment 2:
Materials and Methods: Authors must add the product code of all reagents and kits used in order to allow data reproducibility.
Reply to Reviewer 2 – Comment 2:
We have included the product codes of all relevant reagents and kits. Information was not included for standard chemicals/salts used for buffers (NaCl, TRIS, K2HPO4, Na2HPO4…).
Reviewer 2 - Comment 3:
Figure 2, Immunohistochemistry: Representative images must be shown.
Reply to Reviewer 2 – Comment 3:
We have included the stainings for a control and an eoPE in the initial figure in addition to the already shown loPE.
Reviewer 2 - Comment 4:
Figure 2, Western blots: Representative images must be shown.
Reply to Reviewer 2 – Comment 4:
We have included all Western blots of eoPE, loPE, and controls for all antibodies used in this study.
Reviewer 2-Comment 5:
Supplementary Table 1: this table should be moved in the manuscript and statistically significant differences of the parameters (Age, BMI, Gestational age at delivery...) among the groups must be reported. In addition, in the eoPE group the gestational age is very lower than the control group, this may significantly alter the results obtained by the authors since the expression of the proteins analysed can be dependent on the physiological gestation age rather than the pathology. Thus, an appropriate control group is necessary.
Reply Reviewer 2-Comment 5:
The Supplementary Table 1 was moved into the main manuscript and statistically significant differences of the parameters (Age, BMI, Gestational age at delivery...) among the groups are reported. The association between BMI, age etc as risk factors for preeclampsia are well described. Therefore we included these data in a table. However this is nothing new but is in agreement with numerous published studies.
We appreciate the input from the reviewer. However, healthy fetuses are not delivered preterm under normal circumstances. Almost all indications for preterm delivery are associated with placental pathologies, e.g. premature rupture of membranes, amnion infection, intrauterine growth restriction, and fetal pathology. In all of these cases, the placenta is affected, and vascularization and/or metabolism are disturbed. However, we included a short paragraph noting that the gestational age of the controls must be considered a potential pitfall of the study. Only the results of the group statistics for eoPE could potentially be misinterpreted. The loPE results and the correlations would not be changed by another control group for eoPE.
We included the following statement in the discussion: „One pitfall of the study is the lack of a control group with the same gestational age as the eopE. Since there is no indication to deliver a healthy child prematurely and there is always a pathology behind it, such samples cannot be obtained.“
Reviewer 2-Comment 6:
It is impressive that such a simple study required 13 authors. A detailed author contribution section is necessary.
Reply Reviewer 2-Comment 6:
A detailed author contribution section was included according to the journal guidelines.
Conceptualization, R.G.F., J.A.M. and H.J-B.; methodology, J.A.M. and R.G.F; software, S.H. and T.L.; validation, R.G.F., T.L. and S.H.; formal analysis, T.L., S.H. and R.G.F; investigation, T.L., R.G.F.; resources, N.B., M.B., M. D-P., C.F., D.G., H.J-B.; data curation, T.L., S.H., R.G.F., J.A.M., B.K.; writing—original draft preparation, T.L., R.G.F and H.J-B.; writing—review and editing, J.A.M, B.K, D.W., T.F.; visualization, T.L., R.G.F., and S.H.; supervision, H.J-B. and R.G.F.; project administration, H.J-B.; funding acquisition, R.G.F., H.J-B. All authors have read and agreed to the published version of the manuscript.

Round 2
Reviewer 2 Report
Comments and Suggestions for Authors
the manuscript can be accepted in the current form